# MODEL-BASED ASYNCHRONOUS HYPERPARAMETER AND NEURAL ARCHITECTURE SEARCH

## ABSTRACT

We introduce an asynchronous multi-fidelity method for model-based hyperparameter and neural architecture search that combines the strengths of asynchronous Successive Halving and Gaussian process-based Bayesian optimization. Our method relies on a probabilistic model that can simultaneously reason across hyperparameters and resource levels, and supports decision-making in the presence of pending evaluations. We demonstrate the effectiveness of our method on a wide range of challenging benchmarks, for tabular data, image classification and language modelling, and report substantial speed-ups over current state-of-the-art methods. Our new method, along with asynchronous baselines, are implemented in a distributed framework which will be open sourced along with this publication.

## 1 INTRODUCTION

The goal of hyperparameter and neural architecture search (HNAS) is to automate the process of finding the right architecture or hyperparameters $x_\star \in \arg\min_{x \in \mathcal{X}} f(x)$ of a deep neural network by minimizing the validation loss $f(x)$, observed through noise: $y_i = f(x_i) + \epsilon_i, \quad \epsilon_i \sim \mathcal{N}(0, \sigma^2), i = 1, \ldots, n$. Bayesian optimization (BO) is an effective model-based approach for solving expensive black-box optimization problems (Jones et al., 1998; Shahriari et al., 2016). It constructs a probabilistic surrogate model of the loss function $p(f \mid \mathcal{D})$ based on previous evaluations $\mathcal{D} = \{(x_i, y_i)\}_{i=1}^n$. Searching for the global minimum of $f$ is then driven by trading off exploration in regions of uncertainty and exploitation in regions where the global optimum is likely to reside. However, for HNAS problems, standard BO needs to be augmented in order to remain competitive. For example, training runs for unpromising configurations $x$ can be stopped early, but serve as low-fidelity approximations of $f(x)$ (Swersky et al., 2014; Domhan et al., 2015; Klein et al., 2017b). Further, evaluations of $f$ can be executed in parallel to reduce the wall-clock time required for finding a good solution. Several methods for multi-fidelity and/or distributed BO have been proposed (Kandasamy et al., 2017; 2016; Takeno et al., 2020), but they rely on rather complicated approximations to either select the fidelity level or to compute an information theoretic acquisition function to determine the next candidate. In this work we aim to adopt the desiderata of Falkner et al. (2018), namely, that of simplicity, which often leads to more robust methods in practice.

A simple, easily parallelizable multi-fidelity scheduling algorithm is successive halving (SH) (Karnin et al., 2013; Jamieson & Talwalkar, 2016), which iteratively eliminates poorly performing neural networks over time. Hyperband (Li et al., 2017) iterates over multiple rounds of SH with varying ratios between the number of configurations and the minimum amount of resources spent per configuration. Falkner et al. (2018) introduced a hybrid model, called BOHB, that uses a probabilistic model to guide the search while retaining the efficient any-time performance of Hyperband. However, both SH and BOHB can be bottlenecked due to their *synchronous* nature: stopping decisions are done only after synchronizing all training jobs at certain resource levels (called *rungs*). This approach is wasteful when the evaluation of some configurations take longer than others, as it is often the case when training neural networks (Ying et al., 2019), and can substantially delay progress towards high-performing configurations (see example shown in Figure 1).

Recently, Li et al. (2018) proposed ASHA, which adapts successive halving to the asynchronous parallel case. Even though ASHA only relies on the random sampling of new configurations, it has been shown to outperform synchronous SH and BOHB. In this work, we augment ASHA with a Gaussian process (GP) surrogate, which jointly models the performance across configurations and

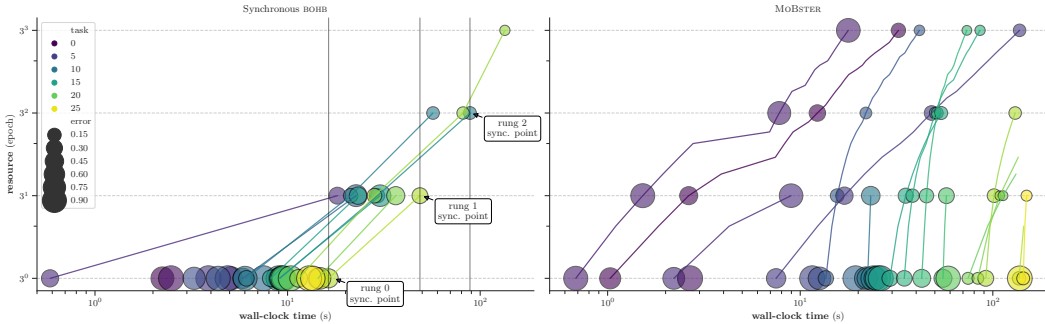

Figure 1: Evolution of synchronous (left) and asynchronous (right) SH for optimization the hyperparameters of a MLP. $x$-axis: wall-clock time (150 secs), $y$-axis: training epochs done (log scale). Coloured lines trace training jobs for HP configurations, with marker size proportional to validation error (smaller is better). Lines not reaching the maximum number of authorized epochs (27) are stopped early. For synchronous SH, rungs (epochs 1, 3, 9) need to be filled completely before any configuration can be promoted, and each synchronization results in some idle time. In asychronous SH, configurations proceed to higher epochs faster. Without synchronization overhead, more configurations are promoted to the highest rung in the same wall-clock time (figure best seen in colours).

rungs to improve the already strong performance of ASHA. The asynchronous nature further requires to handle pending evaluations in principled way to obtain an efficient model-based searcher.

## 1.1 CONTRIBUTIONS

Dealing with hyperparameter optimization (HPO) in general, and HNAS for neural networks in particular, we would like to make the most efficient use of a given parallel computation budget (e.g., number of compute instances) in order to find high-accuracy solutions in the shortest wall-clock time possible. Besides exploiting low-fidelity approximations, we demonstrate that *asynchronous parallel* scheduling is a decisive factor in cost-efficient search. We further handle pending evaluations through fantasizing (Snoek et al., 2012), which is critical for asynchronous searchers. Our novel combination of asynchronous SH with multi-fidelity BO, dubbed MOdel Based aSynchronous mulTi fidelity optimizER (MOBSTER) substantially outperforms either of them in isolation. More specifically:

- We clarify differences between existing asynchronous SH extensions: ASHA, as described by Li et al. (2018) and an arguably simpler stopping rule variant, related to the median rule (Golovin et al., 2017) and first implemented in Ray Tune (Liaw et al., 2018). Although their difference seem subtle, they lead to substantially different behaviour in practice.

- We present an extensive ablation study, comparing and analysing different components for asynchronous HNAS. While these individual components are not novel, one of our main contributions is to show that, by systematically combining them to form MOBSTER, we obtain a reliable and more efficient method than other recently proposed approaches. Due to limited space, we present a detailed description of the technical nuances and complexities of model-based asynchronous multi-fidelity HNAS in Appendix A.3.

- On a variety of neural network benchmarks, we show that MOBSTER is more efficient in terms of wall-clock time than other state-of-the-art algorithms. Unlike BOHB, it does not suffer from substantial synchronization overheads when evaluations are expensive. As a result, we can achieve the same performance in the same amount of wall-clock time but with often just half the computational resources compared to random-sampling based ASHA.

Next, we relate our work to approaches recently published in the literature. In Section 2, we review synchronous SH, as well as two asynchronous extensions. Our novel method is presented in Section 3. We present empirical evaluations for the HNAS of various neural architecture types in Section 4, and finish with conclusions and future work in Section 5.

## 1.2 RELATED WORK

A range of prior Gaussian process (GP) based BO work exploits multiple fidelities of the objective (Kennedy & O'Hagan, 2000; Klein et al., 2017a; Poloczek et al., 2018; Cutajar et al., 2018). A joint GP model across configurations and tasks was used by Swersky et al. (2013), allowing for trade-offs between cheap auxiliaries and the expensive target. Klein et al. (2017a) presented a continuous multi-fidelity BO method, where training set size is an input. It relies on a complicated and expensive acquisition function, and experiments with asynchronous scheduling are not provided.

Kandasamy et al. (2017) presented BOCA, a Bayesian optimization method that exploits general continuous fidelities of the objective function. While BOCA uses a model similar to ours, they employ a different strategy to select the fidelity, which appears to work less efficiently in practice, as we show in our experiments. The "freeze-thaw" approach (Swersky et al., 2014) allows for asynchronous parallel evaluations, yet its scheduling is quite different from asynchronous SH. They propose an exponential decay kernel, a refinement of which we use here. No results for deep neural network tuning were presented, and no implementation is publicly available. Previous work on asynchronous BO has demonstrated performance gains over synchronous BO methods (Alvin et al., 2019; Kandasamy et al., 2016), yet this work did not exploit multiple fidelities of the objective. Concurrently to our work, Takeno et al. (2020) proposed an asynchronous multi-fidelity BO method. However, it is based on an information-theoretic acquisition function that is substantially more complex than our method. Since no implementation is publicly available it is omitted from our comparisons.

Falkner et al. (2018) combine synchronous Hyperband with TPE (Bergstra et al., 2011) as a model. Their method, called BOHB, combines early speed-ups of Hyperband with fast convergence later on, and shows competitive performance for NAS (Ying et al., 2019; Dong & Yang, 2020). However, compared to our approach, it fits independent kernel density estimator at each rung level, which prevents the interpolation to higher rung levels. Apart from this, kernel density estimators do not permit fantasizing the potential outcome of a pending configuration, which makes it unclear how to extend BOHB to the asynchronous setting, and are notoriously sensitive to their bandwidth parameters.

## 2 SYNCHRONOUS AND ASYNCHRONOUS MULTI-FIDELITY SCHEDULING

Multi-fidelity HNAS considers an objective $f(\boldsymbol{x}, r)$, indexed by the *resource level* $r \in \{r_{\min}, \ldots, r_{\max}\}$. The target function of interest is $f(\boldsymbol{x}) = f(\boldsymbol{x}, r_{\max})$, while evaluations of $f(\boldsymbol{x}, r)$ for $r < r_{\max}$ are cheaper, lower-fidelity evaluations that may be correlated with $f(\boldsymbol{x})$. Here, we consider the number of training epochs (*i.e.*, full sweeps over the data) as the resource $r$, though other choices are possible, such as training subset ratios (Klein et al., 2017a).

Two kinds of decisions are made in multi-fidelity HNAS: *choosing* the configurations to evaluate and *scheduling* their evaluations. In this work, configurations are chosen non-uniformly based on the GP surrogate model that accounts for pending and actual evaluations. Scheduling decisions are stop/go decisions that happen at certain resource levels called *rungs*. We make use of successive halving (SH) for its simplicity and strong empirical performance (Jamieson & Talwalkar, 2016; Li et al., 2018). Let $\eta \in \{2, 3, 4\}$ be the halving constant, $r_{\max}$ and $r_{\min}$ the maximum and minimum resource level for an evaluation. We assume, for simplicity, that $r_{\max}/r_{\min} = \eta^K$, where $K \in \mathbb{N}$. The full set of rungs is $\mathcal{R} = \{r_{\min}\eta^k \mid k = 0, \ldots, K\}$. In the sequel, the term "rung" will be overloaded, denoting both the resource level at which stop/go decisions are made and the list of configurations that reached it.

In *synchronous SH*, each rung has an *a priori* fixed size (*i.e.*, number of slots for configurations evaluated until the rung level), the size ratio between successive rungs being $\eta^{-1}$ (e.g., $n(r) = r_{\max}/r$ for $r \in \mathcal{R}$). A round of the algorithm starts with evaluating $n(r_{\min})$ configurations to the lowest rung $r = r_{\min}$, making use of parallel computation if available. Once all of them finish, the $\eta^{-1}$ fraction of top performing configurations are promoted to the next rung while the others are terminated. Each rung is a synchronization point for many training tasks: it has to be populated entirely before any configuration can be promoted (see Figure 1 left for a visualization).

Synchronous scheduling makes less efficient use of parallel computation than asynchronous scheduling. Configurations in the same rung may require quite different compute times, for example, if we

search over hyperparameters that control the network size. Hence, some workers may just be idle at a synchronization point. Evaluations at larger resource levels are observed earlier with asynchronous scheduling. However, the risk is to continue mediocre configurations because they were selected earlier. Figure 1 illustrates the differences between asynchronous and synchronous scheduling.

We distinguish two variants of *asynchronous SH* proposed in prior work. During the optimization process, for any hyperparameter and architecture configuration $x \in \mathcal{X}$ that is currently being evaluated, a decision needs to be made once it reaches the next rung $r \in \mathcal{R}$. Given the new observed data point $y$ at level $r$, the binary predicate continue($x, r, y$) evaluates to true iff $y$ is in the top $\eta^{-1}$ fraction of records at the rung.

**Stopping variant.** This is a simple extension of the median stopping rule (Golovin et al., 2017), which is implemented in Ray Tune (Liaw et al., 2018). As soon as a job for $x$ reaches a rung $r$, if continue($x, r, y$) is true, it continues towards the next rung level. Otherwise, it is *stopped*, and the worker becomes free to pick up a novel evaluation. As long as fewer than $\eta$ configurations have reached a rung, the job is always continued.

**Promotion variant.** This variant of asynchronous SH was presented by Li et al. (2018) and called *ASHA*. Note that, in the remaining text, we will refer to general asynchronous SH as ASHA and indicate the both variants with promotion or stopping based ASHA. Once a job for some $x$ reaches a rung $r$, it is *paused* there, and the worker is released. The evaluation of $x$ can be *promoted* (*i.e.*, continued to the next rung) later on. When a worker becomes available, rungs are scanned in descending order, running a job to promote the first paused $x$ for which continue($x, r, y$) is true. When no such paused configuration exists, a new configuration evaluation is started. With fewer than $\eta$ metrics recorded at a rung, no promotions happen there.

The stopping and promotion variants can exhibit a rather different behaviour (see Figure 2), as also demonstrated in our experiments. Initially, the stopping variant gives most configurations the benefit of doubt, while the promotion variant pauses evaluations until they can be compared against a sufficient number of competitors.

Finally, note that asynchronous SH can be generalized to asynchronous Hyperband in much the same way as in the synchronous case, as shown by Li et al. (2018). However, it was reported that asynchronous SH typically outperforms the Hyperband for expensive tuning problems, which we also show in our experiments in Section 4.1. After some initial exploration, both with random and model-based variants, we confirmed this observation. In the following, we will thus restrict our attention to SH scheduling.

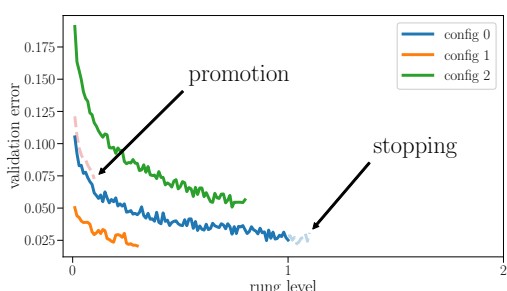

Figure 2: This cartoon on toy data shows the difference between the promotion and stopping variants of SH. The three curves correspond to the evaluations of three configs. The blue curve reaches the smallest rung level first, which triggers the scheduling process. While the stopping variant would continue the evaluation (shaded blue curve), the promotion variant would stop, and proceed with a new configuration (shaded red curve). The evaluation of blue would only be continued once enough datapoints at this rung level have been observed. As shown in the experiments, the stopping variant achieves often better performance in early iterations, whereas the promotion variant obtains usually a better final performance.

## 3 MODEL-BASED ASYNCHRONOUS MULTI-FIDELITY OPTIMIZER

We now describe our new model-based method that is able to make asynchronous decisions across different fidelities of the objective function and in presence of pending evaluations. Motivated by the fact that in standard sequential settings, Gaussian process (GP) based BO outperforms random search (Shahriari et al., 2016), we equip asynchronous SH with decisions based on a *GP surrogate model*. Following a range of prior work on non-parallel multi-fidelity BO (Swersky et al., 2013; Klein et al., 2017a; Poloczek et al., 2018; Swersky et al., 2014), we place a GP prior over $f(x, r)$ to jointly model not only the auto-correlation within each fidelity, but also the cross-correlations between fidelities,

$$f(\boldsymbol{x}, r) \sim \mathcal{GP}(\mu(\boldsymbol{x}, r), k((\boldsymbol{x}, r), (\boldsymbol{x}', r'))).$$

This is in contrast to synchronous BOHB (Falkner et al., 2018), which just employs an independent TPE (Bergstra et al., 2011) model over $\boldsymbol{x}$ for each rung level $r \in \mathcal{R}$, preventing it from transfering information from lower to higher rungs.

**Pending evaluations and fantasizing.** Since multiple configurations are evaluated in parallel, we have to deal with *pending* evaluations (for which results have not yet been obtained) when proposing new configurations. Failing to do so runs the risk of choosing the same (or very similar) configurations over and over, which would result in a poor sampling efficiency. We do so by *fantasizing* the outcomes of pending evaluations, as proposed by Snoek et al. (2012). In a nutshell, we marginalize the acquisition function over the GP predictive posterior for pending configurations. In practice, we approximate this by averaging the acquisition function over sampled function values.

In the context of synchronous scheduling, dealing with pending evaluations is also known as *batch HPO*, a problem which has seen a lot of prior attention (Ginsbourger et al., 2010; Desautels et al., 2014; González et al., 2016; Alvin et al., 2019). Surprisingly, most of these works ignore fantasizing, claiming it is too expensive to do (Ginsbourger et al., 2011), and propose heuristics like local penalties or artificial randomization by Thompson sampling (Hernandez-Lobato et al., 2017; Kandasamy et al., 2016). Quite in contrast to such claims, we find that, as long as kernel hyperparameters remain constant, fantasizing comes almost for free. For GP regression, the posterior covariance matrix does not depend on the function values (only on the inputs), which means that fantasizing does not require additional cubic-complexity computations. Details are given in Appendix A.1.

Lastly, it is important to underscore the unique challenges posed by pending evaluations and fantasizing in the asynchronous SH case, in particular, those not encountered in the synchronous case. We elaborate on these challenges in Appendix A.3 , and summarize our asynchronous algorithm for overcoming them in concise pseudocode.

**Kernel choice.** We implemented and compared a range of different surrogate models for $f(\boldsymbol{x}, r)$: a joint Matérn $5/2$ kernel with automatic relevance determination (ARD) (Snoek et al., 2012) over $(\boldsymbol{x}, r)$, with and without warping of $r$; the additive exponential decay model of Swersky et al. (2014); a novel non-additive extension of the latter, relaxing underlying assumptions which may not hold in practice. In all cases, the base kernel on $\boldsymbol{x}$ alone is Matérn $5/2$ ARD. Details for our new surrogate model are provided in Appendix A.2.

**Acquisition strategy.** We follow synchronous BOHB (Falkner et al., 2018) in choosing $\boldsymbol{x}$ by optimizing the expected improvement (EI) acquisition function (Shahriari et al., 2016) over the posterior on $f(\cdot, r_{\text{acq}})$, where $r_{\text{acq}}$ is fixed for each decision. We choose $r_{\text{acq}} \in \mathcal{R}$ as the largest rung for which at least $L$ metric values have been observed, where $L$ is typically set to the input dimensionality (*i.e.*, number of hyperparameters). After some initial period, $r_{\text{acq}}$ eventually converges to $r_{\max}$. This strategy does not take into account that the choice of $\boldsymbol{x}$ may have different downstream stopping or promotion consequences, interacting with currently pending evaluations. We leave the exploration of alternative acquisition strategies as an important direction for future work.

## 4 EXPERIMENTS

We compare MOBSTER to a range of state-of-the-art algorithms. For synchronous Hyperband and BOHB, we used the `HpBandSter` package.[1], which uses a simple mechanism to reduce the amount of idling workers: if there are more free workers than slots in the rung, left-over workers are used to immediately start the next bracket. There is no open-source implementation of Swersky et al. (2014) and Takeno et al. (2020) for which the implementation complexity is considerable, which is why we omitted them here.

We implemented MOBSTER, as well as all other asynchronous techniques, in `AutoGluon`.[2], which will all be open-sourced along with this paper. A number of worker nodes run training jobs, publishing metrics after each epoch. One of the workers doubles as master, concurrently running the HP optimization and scheduling.

---

[1] https://github.com/automl/HpBandSter
[2] https://github.com/awslabs/autogluon

We also compare against Dragonfly (Kandasamy et al., 2019). While `AutoGluon` allows to parallelize both across instances and processes on a single instance, Dragonfly only supports the latter. Hence, for a fair comparison, we run our method only on a single instance if we compare against Dragonfly. For all other experiments we parallelize across multiple instances.

For all methods using SH the halving rate was $\eta = 3$. We report the *immediate regret* $r_t = y_t - y_\star$ after wall-clock time $t$, where $y_\star$ is the best observed validation error across all methods, runs and time steps on this dataset. Using the regret instead of just the validation error gives a more detailed picture of progress towards the global optimum. For all classification datasets we cap the regret at a lower bound of $10^{-3}$, which corresponds to a $0.1\%$ difference in classification accuracy to the global optimum. We consider lower values to be not significant in practice and attributed them to the intrinsic noise of the objective function. For each method we report mean and the standard error of the mean across multiple runs with different seeds of the random number generator.

We run experiments on the following set of benchmarks (for further details regarding the search spaces see Appendix B. All experiments were conducted on AWS EC2, using `m4.xlarge` instances for CPU- and `g4dn.xlarge` instances for GPU-based computations, respectively.

**Tabular Datasets:** First, we optimized the hyperparameters of a multi-layer perceptron (MLP) on the tabular classification dataset *electricity* gathered from OpenML (Vanschoren et al., 2014). The MLP has two hidden layers with ReLU activations. We tuned 9 hyperparameters: the learning rate of ADAM (Kingma & Ba, 2015), batch size, weight decay, and, for each layer, dropout rate, number of units and the scale of a uniform distribution to initialize weights. The number of epochs $r$ varies between $r_{\min} = 1$ and $r_{\max} = 81$.

**Neural Architecture Search:** To evaluate our approach in a NAS setting, we used the NASBench201 datasets from Dong & Yang (2020) which consists of a offline evaluated grid of neural network architectures for three different datasets: CIFAR-10, CIFAR-100 (Krizhevsky, 2009) and Imagenet16 (Chrabaszcz et al., 2017). Categorical hyperparameters were one-hot encoded. The minimum and maximum number of epochs were $r_{\min} = 1$ and $r_{\max} = 200$. To simulate realistic dynamics of asynchronous NAS, we put each worker to sleep for the time per epoch each real evaluation took. This does not speed up experiments, but decreases cost, since no GPU instances are required. Compared to the other benchmarks which contains integer and continuous hyperparameters, this benchmark is purely discrete.

**Image Classification:** Next, we tuned ResNet (He et al., 2016) on the image classification dataset CIFAR-10 (Krizhevsky, 2009), the batch size, weight decay, initial learning rate and momentum. We used 40000 images for training, 10000 for validation, and applied standard data augmentation (cropping; horizontal flips). SGD is used for training, the learning rate is multiplied by $0.1$ after 25 epochs. We used $r_{\min} = 1$ and $r_{\max} = 27$. As our goal is to evaluate our approach, we omitted additional tricks-of-the-trade required to obtain state-of-the-art accuracy (Yang et al., 2020).

**Language Modelling:** Lastly, we considered an LSTM (Hochreiter & Schmidhuber, 1997), applied to language modelling on the WikiText-2 dataset (Merity et al., 2017) with the default training/validation/test split. We optimized the initial learning rate of SGD, batch size, dropout rate, gradient clipping range, and decay factor multiplied to the learning rate if the validation error has not improved upon the previous epoch. The LSTM consisted of two layers with 1500 hidden units each. For the word embeddings, we also used 1500 hidden units and tied them with the output. Epochs vary between $r_{\min} = 1$ and $r_{\max} = 81$.

### 4.1 ABLATION STUDY

We start with an in-depth analysis of our proposed method. If not explicitly stated otherwise, we report the immediate regret over wall-clock time of the MLP benchmark on the *electricity* dataset for 4 workers and the NASBench201 CIFAR-10 dataset for 8 workers. These two benchmarks cover continuous as well as discrete hyperparameter optimization problems.

**Successive Halving vs Hyperband:** First, we compare asynchronous scheduling based on SH versus Hyperband. We run different instances of SH, where we vary the minimum rung level $r_{min}$ in the same way as Hyperband. Figure 3 shows the results for both promotion and stopping based scheduling, combined with sampling from our model or random. Similar to results reported by Li et al. (2018), the performance of SH is decaying with an increasing $r_{min}$ both for our model-based

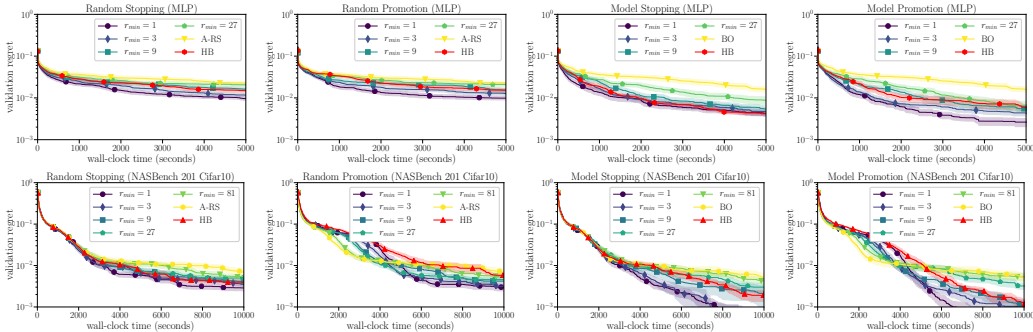

Figure 3: Comparison of asynchronous Hyperband and SH (stopping / promotion scheduling variants and sampling from model / random) on the MLP benchmark (top) and NASBench201 (bottom). SH with the most aggressive bracket works as well and often better than Hyperband across all variants. As comparison we also include asynchronous random search (RS).

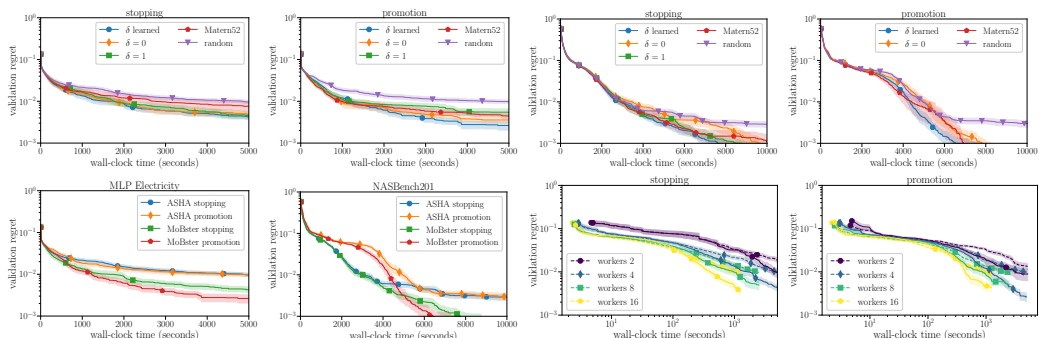

Figure 4: **Top:** Effect of the kernel choice on the MLP benchmark (left, left middle) and NAS-Bench201 (right middle, right). We also include ASHA (random choices) for comparison. There is not a consistent trend across the difference kernels (note the standard error of the mean). **Bottom:** Comparison of ASHA and MOBSTER (stopping and promotion variant). Left: MLP on *electricity* (4 workers). Left Middle: NASBench201 on CIFAR-10 (8 workers). Right Middle: scaling with the number of workers of ASHA (dashed line) and MOBSTER (solid line) for stopping (left) and promotion (right)

and random sampling. Also, the most aggressive bracket, where $r_{min} = 1$ epoch, works as good or even better than Hyperband, most distinctly for the promotion based scheduling. For the remaining experiments we therefore use SH with a minimum rung level of $r_{min} = 1$ epoch.

**Kernels:** We analyse the influence of the choice of kernel and mean function for $f(\boldsymbol{x}, r)$ on the performance of the stopping and promotion variant of asynchronous BOHB. We compare different variants of the exponential decay model (see Appendix A.2. for more details): fixing $\delta = 0$ as in Swersky et al. (2014), $\delta = 1$, or learning $\delta$ with the other GP hyperparameters. We use a Matérn5/2 ARD kernel for the $k_\chi$ model between configurations $\boldsymbol{x}$ Snoek et al. (2012). As an additional baseline, we use a Matérn5/2 ARD kernel and constant mean function on inputs $(\boldsymbol{x}, r)$.

In Figure 4 we compare the various types of kernel. Maybe somewhat surprisingly, there is no significant difference between the surrogate models and the precise model for inter- or extrapolation along $r$ seems less important. For the remaining experiments, we use the exponential decay kernel with learned $\delta$ as the most flexible choice which also obtains slightly better performance.

MOBSTER **vs ASHA:**. Next, we show the benefit of sampling from our model instead of uniformly at random for both scheduling variants (see Figure 4). Initially, MOBSTER (stopping / promotion) performs similar to its random-based counterpart ASHA (stopping / promotion). However, with enough datapoints, it rapidly approaches the global optimum, which is in line with typical improvements of BO over random search (Falkner et al., 2018).

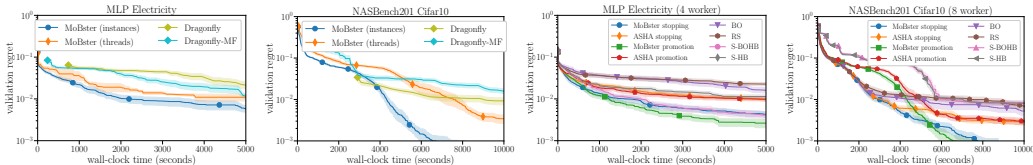

Figure 5: Comparison to various baselines across multiple processes on a single instance (left) and across instances (right) for the MLP and NASBench201 benchmarks. We performed 20 runs for each method. See main text for an analysis.

**Scaling with the number of workers:** As seen in Figure 4 right, the substantial speed-ups of MOBSTER over ASHA are observed consistently with 2, 4, 8, and 16 workers on the MLP benchmark. In fact, in most cases MOBSTER converges to a high-accuracy solution in about the same time as ASHA needs with twice the number of workers: *using a joint surrogate model can save half the resources compared to random search*.

**Stopping versus promotion scheduling:.** Next, we focus on the two variants of asynchronous scheduling (stopping and promotion) described in Section 2. The stopping rule achieves a good performance faster than the promotion variant (Figure 3, right) up to 4000 seconds on the NASBench201 benchmark and up to 500 seconds on the MLP benchmark. Recall from Section 2 that for the stopping rule, the initial $\eta$ configurations are not stopped, while the promotion variant pauses all initial configurations at the lowest rung level of their bracket. However, at least for the MLP benchmark, after this initial phase the promotion-based scheduling achieves slightly better performance.

## 4.2 COMPARISON TO STATE-OF-THE-ART METHODS

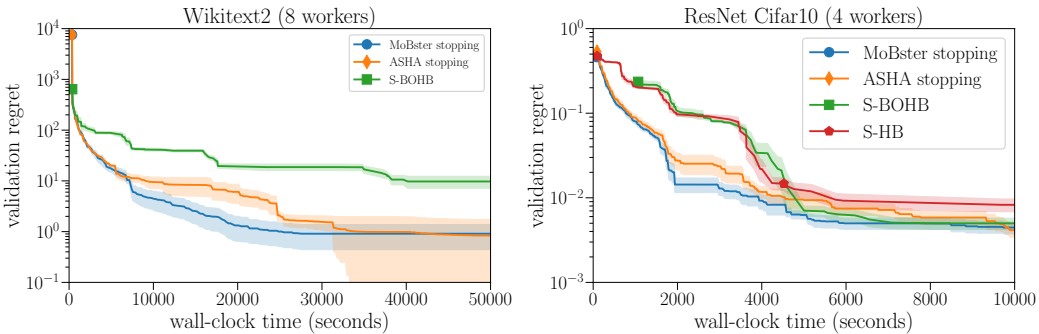

Figure 6: Left: Results of the LSTM benchmark. Right: Results for the ResNet benchmark.

We now compare MOBSTER against a range of state-of-the-art algorithms: ASHA (Li et al., 2018), asynchronous BO (Snoek et al., 2012) (A-BO), asynchronous random search (Bergstra et al., 2011) (A-RS), synchronous BOHB (Falkner et al., 2018) (S-BOHB) , synchronous Hyperband (Li et al., 2017) (S-HB), asynchronous BO with Thompson Sampling (Kandasamy et al., 2016) (Dragonfly) and BOCA (Kandasamy et al., 2017) (Dragonfly-MF).

**MLP and NASBench201** Figure 5 provides results for the MLP and the NASBench201 benchmark, both for the single-instance (left) and multi-instance setting (right) (see Appendix for results on the other two NASBench201 benchmarks). MOBSTER outperforms both Dragonfly and Dragonfly-MF. Thereby, parallelizing across instances provides an additional speed-up compared to the single-instance setting. Somewhat surprisingly Dragonfly-MF is slower in terms of convergence than Dragonfly on the NASBench201 benchmark, which indicates that it cannot model the performance across fidelities well. For the multi-instance scenario, ASHA (stopping and promotion) performs better than synchronous Hyperband and BOHB on the discrete NASBench201 benchmarks. But, due to its random sampling, it often does not converge to a sensible performance on the continuous MLP problem. We attributed S-BOHB poor performance on the NASBench201 datasets to its kernel density estimator which seems to struggle with discrete spaces. As described above, the stopping

variant of MOBSTER works better in the beginning, but the promotion variant achieves a better final performance. However, both variants of MOBSTER outperform all other baselines.

**ResNet on CIFAR-10.** Based on the previous results we limit our comparison to ASHA as well as S-HB and S-BOHB. MOBSTER achieves consistently a lower regret than all three baselines.

**LSTM on WikiText-2.** Figure 6 left shows results for LSTM tuning on the WikiText-2 dataset. To save computations, we limited the comparison to MOBSTER, ASHA, and S-BOHB. Note that a single run (out of 5 per method) takes roughly 5 GPU days. Training a single neural network for the full 81 epochs takes roughly 12 hours, and S-BOHB is only able to execute a single bracket in the allocated time budget. Due to limited data, it cannot fit a surrogate model at the highest resource level $r_{max}$, which may explain its overall poor performance. Because of its asynchronous scheduling, both MOBSTER and ASHA perform substantially better. After around 7000 seconds, MOBSTER's model kicks in, allowing it to approach the optimum much faster than random-based ASHA.

## 5 CONCLUSIONS

We present a novel method, that combines asynchronous scheduling with multi-fidelity BO. Instead of sampling new configurations at random, they are chosen based on a joint Gaussian process surrogate model over resource levels, and asynchronous scheduling is accommodated by fantasizing the outcome of pending candidates. On a range of challenging benchmarks, we compare our proposal to prior synchronous and asynchronous HNAS state-of-the-art. MOBSTER significantly outperforms synchronous BOHB and ASHA. The former is slowed down by synchronization overhead, using its parallel resources inefficiently at higher resource levels. ASHA on the other hand does not suffer from synchronization points, but compared to MOBSTER, we often find it to plateau before solutions with state-of-the-art performance are found.

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

## A  FURTHER DETAILS ABOUT THE MODEL

Here, we provide some additional details in order to support the main text.

### A.1  POSTERIOR REPRESENTATION. FANTASIZING

We employ a standard representation of the Gaussian process posterior. Say that $k(\boldsymbol{x}, \boldsymbol{x}')$ is the kernel function, $\mu(\boldsymbol{x})$ the mean function, and the dataset $\mathcal{D}$ is given by $\boldsymbol{X} \in \mathbb{R}^{n \times d}$, $\boldsymbol{y} \in \mathbb{R}^n$. For notational simplicity, $\boldsymbol{x}$ contains the resource attribute $r$ here. The representation maintains $\boldsymbol{L}$ and $\boldsymbol{p}$, where

$$\boldsymbol{L}\boldsymbol{L}^T = \boldsymbol{K} + \sigma^2 \boldsymbol{I}_n, \quad \boldsymbol{p} = \boldsymbol{L}^{-1}(\boldsymbol{y} - \mu(\boldsymbol{X})).$$

Here, $\boldsymbol{K} = [k(\boldsymbol{x}_i, \boldsymbol{x}_j)] \in \mathbb{R}^{n \times n}$ is the kernel matrix over the training inputs $\boldsymbol{X}$, $\sigma^2$ is the noise variance, and $\boldsymbol{L}$ is the Cholesky factor (lower triangular, positive diagonal).

Recall that we use *fantasizing* (Snoek et al., 2012) to deal with pending feedback. For $M$ fantasy samples, we now have a target *matrix* $\boldsymbol{Y} \in \mathbb{R}^{n \times M}$. For any observed configuration $\boldsymbol{x}_i$, the corresponding row in $\boldsymbol{Y}$ is $y_i \boldsymbol{1}_M^T$. Now, $\boldsymbol{L}$ remains the same, and $\boldsymbol{P} = \boldsymbol{L}^{-1}(\boldsymbol{Y} - \mu(\boldsymbol{X})\boldsymbol{1}_M^T)$. Importantly, the dominating computation of $\boldsymbol{L}$ is independent of the number $M$ of fantasy samples. Predictions on test points result in $M$ different posterior means, but common posterior variances.

In practice, the surrogate model (kernel and mean function) comes with hyperparameters (*e.g.*, noise variance $\sigma^2$, length scales and variance of kernel $k_{\mathcal{X}}$, $\alpha, \beta, \gamma, \delta$), which have to be adjusted. All in all, recomputing the posterior representation involves (a) optimizing the hyperparameters on the observed dataset only, and (b) drawing fantasy targets, then computing the representation on the extended dataset, which includes pending configurations. Here, (a) is substantially more expensive than (b), as it involves computing the representation many times during the optimization.

In our implementation, we delay computations of the posterior representation until the next configuration has to be chosen. Our algorithm is later formalized by concise pseudocode in appendix A.4. Note that if running tasks report metric values, they are inserted into the dataset, replacing pending configurations entered previously. Our implementation offers a number of strategies to skip the expensive part (a) of the computation. Here, (a) is not skipped initially, until the dataset size passes a threshold. After that, we can skip (a) for all but every $k$-th update. Another strategy is to skip (a) as long as the number of *full resource* datapoints $(\boldsymbol{x}_i, r_i = r_{\max})$ does not grow. This ensures that we do not spend more time on surrogate model updates in the multi-fidelity case than in standard BO with full resource evaluations only.

### A.2  EXPONENTIAL DECAY SURROGATE MODEL

The model proposed by Swersky et al. (2014) is based on the assumption $f(\boldsymbol{x}, r) = f(\boldsymbol{x}) + e^{-\lambda r}$, where $\lambda \sim p(\lambda)$ is drawn from a Gamma prior, and the asymptote $f(\boldsymbol{x})$ (*i.e.*, $f(\boldsymbol{x}, r)$ as $r \to \infty$) from a GP prior. They end up using a zero-mean GP prior with kernel function $k((\boldsymbol{x}, r), (\boldsymbol{x}', r')) = k_{\mathcal{X}}(\boldsymbol{x}, \boldsymbol{x}') + k_{\mathcal{R}}(r, r')$ and

$$k_{\mathcal{R}}(r, r') := \int_0^\infty e^{-\lambda r} e^{-\lambda r'} p(\lambda) \, d\lambda = \kappa(r + r'),$$

$$\kappa(u) := \frac{\beta^\alpha}{(u + \beta)^\alpha}, \quad \alpha, \beta > 0.$$

This proposal has several shortcomings. First, $f(\boldsymbol{x}, r) \to f(\boldsymbol{x}) + 1$ as $r \to 0$, independent of what metric is used. Second, the random function $e^{-\lambda r}$ assumption implies a non-zero mean function, and a kernel taking this into account. A better assumption would be $f(\boldsymbol{x}, r) = f(\boldsymbol{x}) + \gamma e^{-\lambda r}$, $\gamma > 0$, which implies a mean function $\mu(\boldsymbol{x}, r) = \mu_{\mathcal{X}}(\boldsymbol{x}) + \gamma \kappa(r)$ and a covariance function

$$k((\boldsymbol{x}, r), (\boldsymbol{x}', r')) = k_{\mathcal{X}}(\boldsymbol{x}, \boldsymbol{x}') + \gamma^2 \tilde{k}_{\mathcal{R}}(r, r'), \tag{1}$$

$$\tilde{k}_{\mathcal{R}}(r, r') := \kappa(r + r') - \kappa(r)\kappa(r').$$

One advantage of this additive model structure is that an implied conditional independence relation can be exploited in order to speed up inference computations (Swersky et al., 2014). However, this model implies $f(\boldsymbol{x}, r) = f(\boldsymbol{x}) + \gamma$ as $r \to 0$: the metric for no training at all depends on $\boldsymbol{x}$ in the

same way as the asymptote. A more sensible assumption would be $f(\boldsymbol{x}, r) = \gamma$ as $r \to 0$, since without training, predictions should be random (*e.g.*, $\gamma = \frac{1}{2}$ for binary classification). We present a novel surrogate model which can incorporate this assumption. It is based on

$$f(\boldsymbol{x}, r) = \gamma e^{-\lambda r} + f(\boldsymbol{x}) \left(1 - \delta e^{-\lambda r}\right), \quad \delta \in [0, 1]. \tag{2}$$

Some algebra (provided below) gives

$$\mu(\boldsymbol{x}, r) = \gamma \kappa(r) + \mu_{\mathcal{X}}(\boldsymbol{x})(1 - \delta\kappa(r)),$$
$$k((\boldsymbol{x}, r), (\boldsymbol{x}', r')) = (\gamma - \delta\mu_{\mathcal{X}}(\boldsymbol{x}))\tilde{k}_{\mathcal{R}}(r, r')(\gamma - \delta\mu_{\mathcal{X}}(\boldsymbol{x}'))$$
$$+ k_{\mathcal{X}}(\boldsymbol{x}, \boldsymbol{x}')[1 - \delta(\kappa(r) + \kappa(r')$$
$$- \delta\kappa(r + r'))].$$

For $\delta = 0$, we recover the additive model of Eq. 1, while for $\delta = 1$, we have that $f(\boldsymbol{x}, r) = \gamma$ as $r \to 0$, independent of $\boldsymbol{x}$. This model comes with hyperparameters $\alpha, \beta, \gamma > 0$ and $\delta \in [0, 1]$. We provide an empirical analysis of this model in Section 4.1 of the main paper.

### DERIVATION OF EXPONENTIAL DECAY SURROGATE MODEL

Recall that our novel surrogate model $k((\boldsymbol{x}, r), (\boldsymbol{x}', r'))$ and $\mu(\boldsymbol{x}, r)$ is based on the random function assumption (2). Since $\mathbb{E}[e^{-\lambda r}] = \kappa(r)$, it is clear that $\mathbb{E}[f(\boldsymbol{x}, r)] = \gamma\kappa(r) + \mu_{\mathcal{X}}(\boldsymbol{x})(1 - \delta\kappa(r)) = \mu_{\mathcal{X}}(\boldsymbol{x}) + (\gamma - \delta\mu_{\mathcal{X}}(\boldsymbol{x}))\kappa(r)$. Moreover,

$$k((\boldsymbol{x}, r), (\boldsymbol{x}', r')) = \mathbb{E}\left[f(\boldsymbol{x}, r)f(\boldsymbol{x}', r')\right]$$
$$= (\gamma - \delta\mu_{\mathcal{X}}(\boldsymbol{x}))\tilde{k}_{\mathcal{R}}(r, r')(\gamma - \delta\mu_{\mathcal{X}}(\boldsymbol{x}')) + k_{\mathcal{X}}(\boldsymbol{x}, \boldsymbol{x}')$$
$$\cdot \mathbb{E}\left[\left(1 - \delta e^{-\lambda r}\right)\left(1 - \delta e^{-\lambda r'}\right)\right] = (\gamma - \delta\mu_{\mathcal{X}}(\boldsymbol{x}))$$
$$\cdot \tilde{k}_{\mathcal{R}}(r, r')(\gamma - \delta\mu_{\mathcal{X}}(\boldsymbol{x}')) + k_{\mathcal{X}}(\boldsymbol{x}, \boldsymbol{x}')$$
$$\cdot \left(1 - \delta(\kappa(r) + \kappa(r') - \delta\kappa(r + r'))\right),$$
$$\tilde{k}_{\mathcal{R}}(r, r') = \kappa(r + r') - \kappa(r)\kappa(r').$$

### A.3 FANTASIZING, SYNCHRONOUS VS. ASYNCHRONOUS

Here we underscore the unique challenges posed by pending evaluations and fantasizing in the asynchronous SH case, in particular, those not encountered in the synchronous case. Specifically, in the asynchronous case, we are offered far fewer guarantees about the algorithmic state at a given point in time, not least when a decision needs to be made about what configuration to propose. This is exacerbated by the fact that concurrent state updates are now possible, making it challenging to incorporate the most up-to-date information in the decision-making process.

In short, to combine asynchronous SH and GP-based BO with fantasizing, there are many more moving parts to contend with than in the synchronous counterpart, and an appreciable amount of care and attention toward bookkeeping is required to make it work efficiently. We illustrate this with a concrete example. Consider an instantiation of SH with $r_{\min} = 1, r_{\max} = 27$ with $\eta = 3$. Further, assume there are a total of 5 workers.

**Synchronous.** A snapshot of the algorithmic state is illustrated in Figure 7, showing the *base rung* of the *first bracket* being filled: so far 6 configurations have been proposed and 2 have been evaluated $(\boldsymbol{x}_1, \boldsymbol{x}_3)$, while 4 are still pending $(\boldsymbol{x}_2, \boldsymbol{x}_4, \boldsymbol{x}_5, \boldsymbol{x}_6)$. This means there is now 1 worker idle, namely, `worker 5`. Since synchronous SH dictates that we need to completely evaluate 27 configurations at this rung before deciding which 9 to promote to the next rung, we still have 21 configurations to propose and evaluate. Hence, the scheduler will assign `worker 5` to evaluate a new configuration at the base rung.

Proposing the new configuration in the presence of these pending evaluations is straightforward. One a) updates the GP model with all completed evaluations, and then b) sample fantasy outcomes for the pending evaluations. Note that one need only extrapolate to the current rung level, so little to no bookkeeping is required. Thereafter, one simply c) optimizes the acquisition function, conditioned on observed and fantasized evaluations.

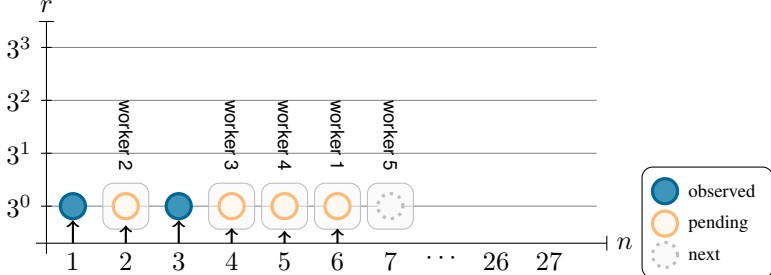

Figure 7: Synchronous. Base rung of the bracket has yet to be filled (6/27). Therefore, the idle worker will be assigned to evaluate a new configuration at this rung. Proposing this new configuration with GP-BO is straightforward, even in the presence of pending evaluations, since they are all pending at the same rung.

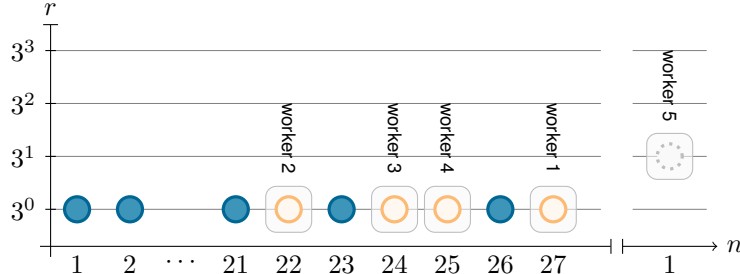

Figure 8: Synchronous, edge case. Base rung of the bracket is soon to be filled (23/27). With 4 workers assigned to evaluate the last 4 configurations necessary to fill this rung, the idle worker will be assigned to evaluating a new configuration at the base rung of the next bracket. Again, all pending evaluations are pending at the same rung.

In the edge case where the configurations necessary to fill the base rung have already been proposed but some are still pending evaluation, a situation might arise where a worker becomes idle but cannot begin evaluations at the next rung until all evaluations are complete at the current rung. An example of this is illustrated in Figure 8. A total of 23 configurations have been evaluated and only 4 configurations are still pending ($x_{22}, x_{24}, x_{25}, x_{27}$). This leaves 1 worker idle. A thoughtful implementation might utilize idle workers to begin filling the base rung of subsequent brackets. Even in this case, proposing new configurations in the presence of these pending evaluations in previous brackets is still relatively straightforward, since they are all pending at the same rung level.

**Asynchronous.** By contrast, in the asynchronous paradigm, there are no guarantees concerning the rungs at which configurations will be pending evaluation.

Furthermore, in an asynchronous computing environment, it is possible that during the scheduling procedure, i.e. deciding whether promote an existing configuration to the next rung or to propose a new configuration to fill the base rung (which would then invoke configuration proposal procedure), it is possible for important events to take place concurrently, not least of which being the completion of evaluations. To simply ignore such events would be reckless, as this could risk wasting resources on evaluating suboptimal configurations proposed with stale data.

A snapshot of an example algorithmic state is illustrated in Figure 9. This snapshot has captured a situation where there is 1 idle worker, and the scheduler has decided to utilize it to evaluate a new configuration in order to fill the base rung. Now it needs to invoke the configuration proposal procedure.

Unlike before, this procedure needs to account for evaluations that have been observed, now across every rung, and deal with evaluations that are still pending, now also possible at every rung. Added to these complexities is the fact that additional evaluations have completed since the scheduling procedure began, namely, those of configurations $x_4$ and $x_9$, at rungs 0 and 2, respectively.

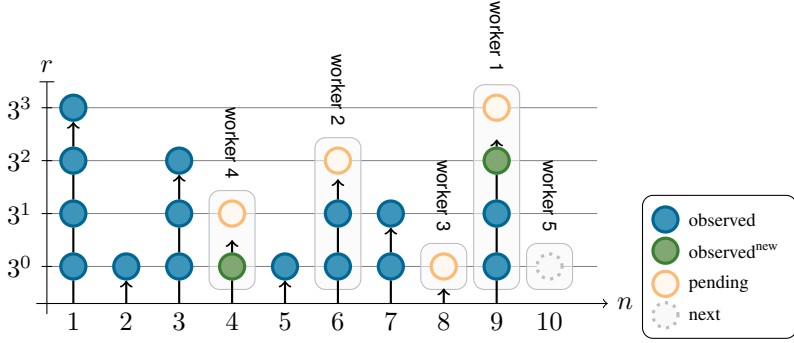

Figure 9: Asynchronous.

To respond to these added complexities and nuances, additional care needs to be taken with algorithmic bookkeeping. In appendix A.4, we describe the space of possible algorithmic states and formalize our asynchronous algorithm for dealing with these added complexities in concise pseudocode.

## A.4 ALGORITHM

First, we formalize some concepts. A *task* is a unit of work for evaluating the blackbox function at a configuration $x$ to some specified rung $r$. Then, it is the responsibility of the *workers* to asynchronously execute these units of work. Each task contains contextual information that includes hyperparameter configuration $x$, the rung $r$, the target value $y$ and a *status* flag indicating the status of the target value. Specifically, the status flag

$$\text{status} \in \{\text{observed}, \text{observed}^{\text{new}}, \text{pending}, \text{fantasized}\}$$

indicates whether an *actual* evaluation has been completed, in which case,

- if it has already been incorporated in updating the surrogate model, then $\text{status} = \text{observed}$,
- otherwise, $\text{status} = \text{observed}^{\text{new}}$.

If the evaluation is currently pending, in which case

- $y$ may have a fantasized value, and $\text{status} = \text{fantasized}$, or
- no value at all, and $\text{status} = \text{pending}$.

Note that, in general, each hyperparameter configuration $x$ will have one or more associated tasks, each with different rungs $r$. Let $\mathcal{D}^{\dagger}$ denote the dataset consisting of all the observed data,

$$\mathcal{D}^{\dagger} = \{(\text{task}.x, \text{task}.r, \text{task}.y) \mid \text{task} \in \text{Tasks} : \text{task}.\text{status} = \text{observed}\},$$

and $\mathcal{D}^{\ddagger}$ denote the *extended* dataset consisting of both observed and fantasized data,

$$\mathcal{D}^{\ddagger} = \{(\text{task}.x, \text{task}.r, \text{task}.y) \mid \text{task} \in \text{Tasks} : \text{task}.\text{status} \in \{\text{observed}, \text{fantasized}\}\}.$$

Our method is summarized in algorithm 1.

---

**Algorithm 1:** MOBSTER

---

1 **while** *under budget* **do**
2   **if** $\exists$ worker $\in$ Workers : worker.status $=$ idle **then**
3     task $\leftarrow$ CreateTask()         // determines the task's configuration $x$ and rung $r$
4     task.$y \leftarrow$ NIL         // initialize target value
5     task.status $\leftarrow$ pending
6     **async call** Execute(task)
7   **end**
8 **end**
9 **Function** CreateTask():
10   Provided in [Li et al. 2020; Algorithm 2] as function get_job(). MOBSTER overrides the line *"Draw random configuration"* with a call to ProposeCandidate()
11 **end**
12 **Function** Execute(task):
13   task.$y \leftarrow f($task.$x,$ task.$r)$         // evaluate $f$ at $x$ with resource $r$
14   task.status $\leftarrow$ observed$^{\text{new}}$
15 **end**
16 **Function** ProposeCandidate(*arguments*):
    **Result:** candidate $x_\star$
17   /* if there are new observations, we need to update the surrogate model and invalidate existing fantasy values   */
18   **if** $\exists$ task $\in$ Tasks : task.status $=$ observed$^{\text{new}}$ **then**
19     **for** task $\in$ Tasks **do**
20       **if** task.status $=$ observed$^{\text{new}}$ **then**         // mark as incorporated
21         task.status $\leftarrow$ observed
22       **else if** task.status $=$ fantasized **then**         // invalidate all fantasy values
23         task.$y \leftarrow$ NIL         // reset target value
24         task.status $\leftarrow$ pending
25       **end**
26     **end**
27     /* maximize surrogate model hyperparameters $\theta$, conditioned on *observed* data $\mathcal{D}^\dagger$   */
28     $\theta_\star \leftarrow \arg\min_{\theta} \mathcal{L}(\theta; \mathcal{D}^\dagger)$
29   **end**
30   **for** task $\in$ Tasks **do**         // impute fantasy values for all pending candidates
31     **if** task.status $=$ pending **then**
32       /* sample from the posterior predictive, conditioned on *observed* data $\mathcal{D}^\dagger$   */
33       task.$y \leftarrow \tilde{y}$, where $\tilde{y} \sim p(y \,|\, $task.$x,$ task.$r; \theta_\star, \mathcal{D}^\dagger)$
34       task.status $\leftarrow$ fantasized
35     **end**
36   **end**
37   /* maximize acquisition function, conditioned on both *observed* and *fantasized* data $\mathcal{D}^\ddagger$   */
38   $x_\star \leftarrow \arg\max_{x \in \mathcal{X}} \alpha(x, r_{\text{acq}}; \theta_\star, \mathcal{D}^\ddagger)$
39 **end**

---

## B  Configuration Spaces

Table 1 shows the configuration space for the MLP benchmark, Table 2 for the residual network benchmark, Table 4 for NASBench201 and Table 3 for the LSTM benchmark. If we use an exponential notation than we optimized the corresponding hyperparameter on a logarithmic scale. We used a one-hot encoding for all categorical hyperparameters and an integer encoding for ordinal hyperparameters.

## C  Additional NASBench201 Experiments

Table 1: The hyperparameters and architecture choices for the fully connected networks.

| Hyperparameter | Range |
|---|---|
| learning rate | $[10^{-6}, 1]$ |
| batch size | $[2^3, 2^7]$ |
| weight decay | $[10^{-8}, 1]$ |
| dropout layer 1 | $[0, .99]$ |
| dropout layer 2 | $[0, .99]$ |
| units layer 1 | $[2^4, 2^{10}]$ |
| units layer 2 | $[2^4, 2^{10}]$ |
| scale layer 1 | $[10^{-3}, 10]$ |
| scale layer 2 | $[10^{-3}, 10]$ |

Table 2: The configuration space for the residual neural network benchmark.

| Hyperparameter | Range |
|---|---|
| learning rate | $[1^{-3}, 1^{-1}]$ |
| batch size | $[8, 256]$ |
| weight decay | $[10^{-5}, 10^{-3}]$ |
| momentum | $[0, .99]$ |

Table 3: The configuration space for the LSTM benchmark.

| Hyperparameter | Range |
|---|---|
| learning rate | $[1, 50]$ |
| batch size | $[2^3, 2^7]$ |
| dropout | $[0, 0.99]$ |
| gradient clipping | $[0.1, 2]$ |
| learning rate factor | $[1, 100]$ |

Table 4: The configuration space for NASBench201.

| Hyperparameter | Range |
|---|---|
| edge 0 | $\{none, skip\text{-}connect, nor\text{-}conv\text{-}1x1, nor\text{-}conv\text{-}3x3, avg\text{-}pool\text{-}3x3\}$ |
| edge 1 | $\{none, skip\text{-}connect, nor\text{-}conv\text{-}1x1, nor\text{-}conv\text{-}3x3, avg\text{-}pool\text{-}3x3\}$ |
| edge 2 | $\{none, skip\text{-}connect, nor\text{-}conv\text{-}1x1, nor\text{-}conv\text{-}3x3, avg\text{-}pool\text{-}3x3\}$ |
| edge 3 | $\{none, skip\text{-}connect, nor\text{-}conv\text{-}1x1, nor\text{-}conv\text{-}3x3, avg\text{-}pool\text{-}3x3\}$ |
| edge 4 | $\{none, skip\text{-}connect, nor\text{-}conv\text{-}1x1, nor\text{-}conv\text{-}3x3, avg\text{-}pool\text{-}3x3\}$ |
| edge 5 | $\{none, skip\text{-}connect, nor\text{-}conv\text{-}1x1, nor\text{-}conv\text{-}3x3, avg\text{-}pool\text{-}3x3\}$ |

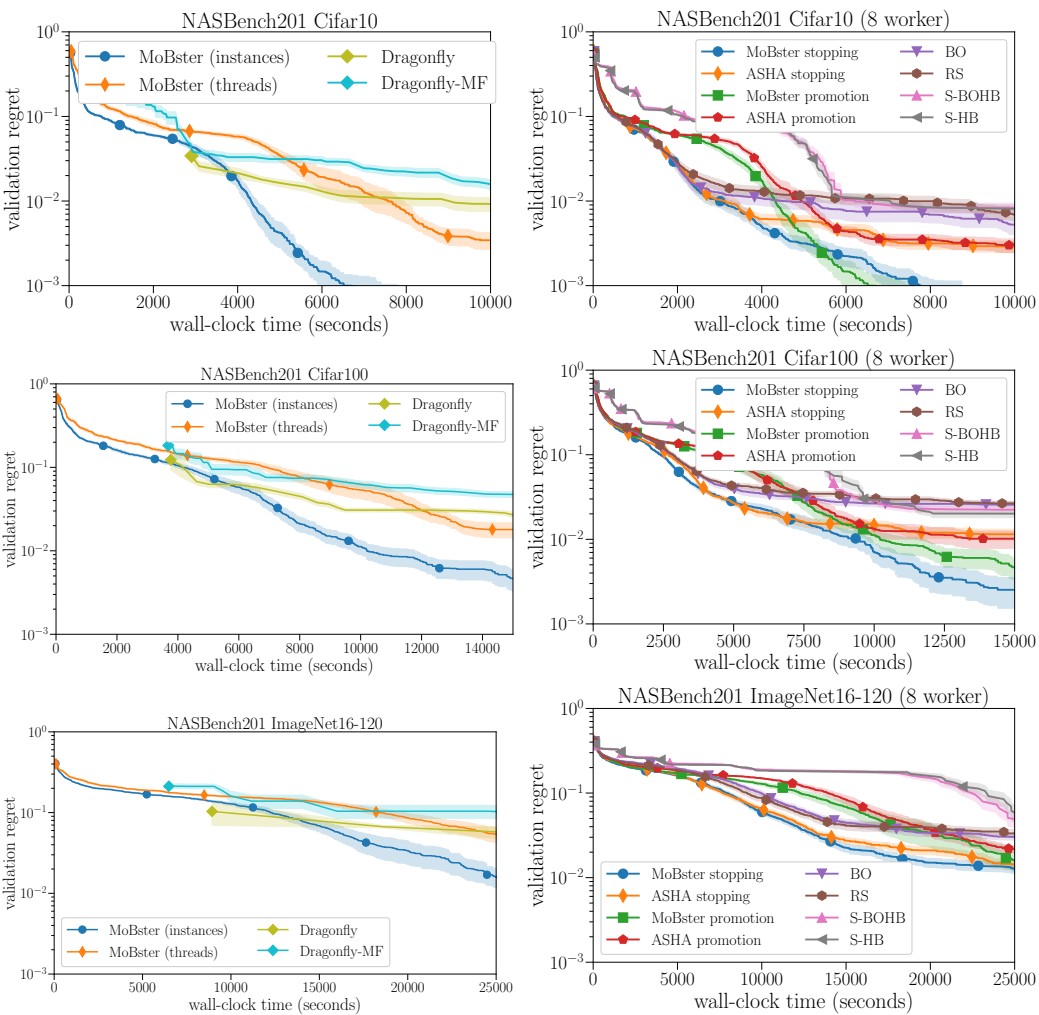

Figure 10: Comparison to various baselines across multiple processes on a single instance (left) and across instances (right) for the three different NASBench201 benchmarks. See main text for an analysis.

