# OpenReview forum: "Model-based Asynchronous Hyperparameter and Neural Architecture Search"
_ICLR.cc/2021/Conference — Reject_

### Official Review · AnonReviewer4 · 2020-10-27
**Good paper with a natural idea of extending GP-based SH to the asynchronous setting, supported by extensive experiments.**

**Rating:** 6
**Confidence:** 4

**Review:**

After rebuttal: First of all, I would like to thank the authors for all their effort on the rebuttal and the revised paper and I really appreciate that. After carefull discussion with AC and other reviewers, I would, however, have to decrease my score to 6 due the lack of significant technical novelty.

-------------------------------------------------------------------------------------------------------------------------------------------------------------------------------

Overview: This paper proposes to combine GP to SH/random sampling for asynchronous hyper-parameter optimization, and it proposes to use scheduling schemes like in ASHA for example (promotion scheduling) to handle the idle workers. The GP is placed over the function f(x, r), thus takes into account the cross-correlations between different fidelities. Ablation studies are provided to show how to choose the minimum rung level, the kernel hyper-parameters, as well as the impact of different number of workers and different scheduling scheme. The benefits of the new algorithm is then proved by extensive experiments to compare the proposed algorithm MobSter against the state-of-the-art algorithms.

Reasons for the score:
I am pretty satisfied with the whole structure and pretty enjoyed reading the paper. The idea of replacing the random sampling in ASHA by a GP looks quite natural and is shown to be promising with thorough ablation studies and extensive comparisons to SOTA algorithms. Therefore, I would like to vote for accept. The only potential default is maybe the fact that all techniques used to form MobSter are not novel individually, which is also recognized by the authors. But I do think the paper is an useful work for the AutoML community.

General questions and remarks: As I said, I'm generally satisfied, and only have a few small remarks.
- I think it would probably be better that the authors put a bit more effort on explaining the notion of rung and bracket (in Section 2). I'm not sure if it is that clear for those who are not familiar with the litterature.
- One thing that I'm not quite convinced is the NAS part. Have the authors considered of comparing MobSter to other NAS-specific algorithms?

Minor comments:
- Section 1.1: one of our main contribution -> contributions.
- Section 4, paragraph 1: for which the complexity considerable -> for which the complexity is considerable.
- Section 4, paragraph 1: which is why omitted them here -> I would rather say 'which is why they are omitted here' probably?
- Section 4, paragraph 4: immediate regret -> I would probably call it dynamic immediate regret, but it's not important. Btw, I'm not sure that I'm fully convinced by the claim "Using the regret instead of just the validation error gives a more detailed picture of progress towards the global optimum.".

---

> ### Author Response · Authors · 2020-11-18
> **Response to reviewer 4**
>
> We thank the reviewer for the encouraging and thorough review and are happy that they enjoyed reading the paper. We agree that the proposed method is a useful contribution to the AutoML community. All minor comments have been resolved in the updated version of the paper.
>
> - Regarding the notion of rung and bracket in Section 2: Thank you for pointing this out to us. We revised this Section in the updated version of our paper.
>
> - We indeed considered to compare against other typical NAS methods, such as DARTS, however these can not be run asynchronously and, since the focus of our paper is the distributed parallel setting, we found this to be an unfair, and perhaps even somewhat distracting, comparison.

---

### Official Review · AnonReviewer3 · 2020-10-27
**Extensive, compelling experimental results but minimal scientific merit**

**Rating:** 5
**Confidence:** 5

**Review:**

This paper has proposed to exploit a GP model to represent the correlation between configuration-rung tuples (as is typical in multi-fidelity BO) in asynchronous successive halving (ASHA) (Li et al. 2018), which has resulted in performance improvement over the state of the art, as shown in the experimental results. The experimental results are extensive and compelling. Introducing a GP to model the correlation between configuration-rung tuples in the existing ASHA work offers minimal technical merit though. Can the authors elaborate on whether there is any nontrivial, novel technical challenge with such an integration? This question does not seem to be adequately addressed in the paper.

Section 3 is missing an algorithm showing the exact technical details of how to tie all the various components mentioned in the paper together. In its present state, it is an amalgamation of several bits and pieces describing the components in words.

Page 1: The authors say that "they rely on rather complicated mechanisms to select either the fidelity level or the next candidate. In this work we aim to adopt the desiderata of Falkner et al. (2018), namely, that of simplicity, which often leads to more robust methods in practice." Can the authors clarify whether they are claiming that multi-fidelity BO or BO with early stopping algorithms (including the references below for which codes are provided for empirical comparison) are less robust due to their complicated mechanisms? If so, can the authors support this claim?

Bayesian optimization meets Bayesian optimal stopping. In ICML 2019.

Information-based multi-fidelity Bayesian optimization. In NeurIPS Workshop on Bayesian Optimization 2017.

Bayesian optimization with binary auxiliary information. In UAI 2020.

The authors say that "This is in contrast to synchronous BOHB (Falkner et al., 2018), which just employs an independent TPE (Bergstra et al., 2011b) model over x for each rung level, which prevents it to transfer information from lower to higher rungs." Can't it be simply extended to cover all rungs?

The authors say that "For each method we report mean and the standard error of the mean across multiple runs with different seeds of the random number generator." Exactly how many runs would that be?



Minor issues

The title, which includes neural architecture search, may not be appropriate, considering that the methodology in Section 3 is not about NAS. A more general title ought to be considered since the proposed methodology is applicable to general BO problems, as seen in the experiments section.

Page 2. substantially behaviours?

Page 3. I would prefer that the authors do not overload the term "rung" to have two very different meanings, considering that it is already not easy to remember what it meant originally.

Page 8. AHSA

Page 9. The following re4ference appears twice: K. Kandasamy, A. Krishnamurthy, J. Schneider, and B. Poczos. Parallelised Bayesian optimisation via Thompson sampling. In Proceedings of the 21st International Conference on Artificial Intelligence and Statistics (AISTATS'18).

---

> ### Author Response · Authors · 2020-11-18
> **Response to reviewer 3**
>
> We thank the reviewer for taking their time to provide a review of our paper. We are happy that the reviewer found the experiments extensive and compelling. Regarding the novelty aspect, we refer to the response above. All minor issues are fixed in the updated version of the paper.
>
> - Regarding the nontrivial, novel technical challenge: There are numerous technical challenges involved in devising a practical asynchronous multi-fidelity method. Not least of these includes fantasizing, which in the asynchronous paradigm is not so straightforward (details of this are elaborated in the Appendix of the revised version of the paper). Apart from that, we are the first to show that while different variants of ASHA (promotion vs stopping) might seem superficial, they have a strong effect on anytime performance.
>
> - “Section 3 is missing an algorithm”: We thank the reviewer for the suggestion and will add a pseudo-code block that summarizes how the different parts of our algorithm interact with each other.
>
> - Regarding extending the TPE model across all rung levels: TPE fits a kernel density estimator to model the density l(x) of the top performing configuration (where yi < tau) and the density g(x) of the poorly performing configurations (where yi > tau). First of all, since it does not model the function directly, it would not be able to differentiate  between hyperparameter configurations x and the resource level r and, hence, it is not able to extrapolate the performance of x across different r. Secondly, this could be too exploitative, particularly in cases where the performance along r varies more than along x (e.g wikitext2 benchmark), which means that l(x) would be clustered on high resource levels ignoring the actual hyperparameters x.
>
> - How many runs per method: We perform 20 runs per method on the MLP and Nasbench201 benchmark, 10 for the ResNet benchmark and 5 for the LSTM benchmark. We will clarify this in the updated version of our paper.
>
> - “Can the authors clarify whether they are claiming that multi-fidelity BO or BO with early stopping algorithms (including the references below for which codes are provided for empirical comparison) are less robust due to their complicated mechanisms?” Several multi-fidelity for HPO have been proposed, for example Fabolas (Klein et al) or Freeze-Thaw (Swersky et al). However, these methods use entropy search to select the configuration x and the fidelity level. While this in theory is more flexible than using a fixed schedule, such as successive halving, it comes with the caveat that they rely on rather complicated approximations, such as modelling the distribution of the global optimum p(x=x_star) to compute the information gain. However, based on the strong empirical results of successive halving (see Falkner et al, Li et al), information theoretic based selection criterion might not be preferable compared to the much simpler successive halving schedule.

---

### Official Review · AnonReviewer1 · 2020-10-28
**Official Blind Review #1**

**Rating:** 6
**Confidence:** 3

**Review:**

##########################################################################

Summary:
The paper proposes a model-based asynchronous multi-fidelity method to optimize hyperparameters and perform a neural architecture search (NAS). The paper begins by addressing the differences between synchronous and asynchronous scheduling in Successive Halvining (SH) and its variants. It also analyzes the different stopping rule criteria. Based on these insights, the paper proposes a unified approach called BOBSTER that combines asynchronous scheduling and multi-fidelity Bayesian optimization. The authors empirically demonstrate that BOBSTER can find the optimal hyperparameters (architecture) more efficiently in terms of wall-clock time than other state-of-the-art algorithms on NAS, image classification task, and language modelling task.

##########################################################################

Pros:
1. The paper presents a simple strategy to speed up hyperparameter optimization / NAS that gives good empirical results. The work is certainly relevant to this venue.
2. The paper is well written and motivated. The release of the codebase can potentially help the research community.
3. The empirical results are convincing and support the claims made in the paper. The experiment protocols and choice of baselines are appropriate.

##########################################################################

Cons:
The key concern about the paper is the lack of novelty. The paper does not bring additional insights. Both asynchronous scheduling and non-parallel multi-fidelity BO were studied before, and the work is an extension of these approaches. The method proposed in the paper looks more like engineering work. Moreover, although the justification is the appendix looks correct, it could be written more precisely (in terms of notations and explanation).

##########################################################################

Questions:
1. What are some limitations of using asynchronous scheduling (e.g. memory?). It would be helpful to discuss these in the manuscript.
2. How accurate is approximating the marginal acquisition function by averaging the acquisition function over sampled function values? How does this affect the model?
3. How does not considering that the choice of x may have different downstream stopping or promotion consequences affect the performance?
4. In figure 4, the results don't show how ASHA changes as a function of the number of workers, but the main text describes it.
5. It seems like validation regret is still decreasing in figures 5 and 6. What happens if it trains longer?
6. What are meta-parameters to BOBSTER, and how sensitive are they? It would be helpful to have a sensitivity study. Also, how does the BOBSTER scale to the number of hyperparameters compared to the baselines? It will be helpful if the paper discusses these points.

Minor notes:
* In figure 1 caption, x, y --> $x$, $y$.
* On page 2, "they lead to substantially behaviours in practice" --> "they lead to substantially different behaviours in practice"? Also, please re-write the previous sentence.
* On page 2, the last bullet point, does the random sampling refer to BOHB? Also, are half the computational resources consistent throughout all experiments?
* On page 3, K is not defined.
* Figure 2 is difficult to understand at a glance. One suggestion is to use a side-by-side comparison between promotion and stopping instead of combining them in the same plot.
* On page 5, how many random seeds are used when reporting the mean and standard error?
* On page 6, ".All" --> ". All"; missing space.
* Many figures in the paper do not show low validation regret; it cuts the curves. Are there any reasons for cutting them?
* On page 6, "no GPU instances are required" --> doesn't the evaluation stage also requires GPU?
* Labels in figures are small and hard to read (especially figure 5).
* If these ASHA type methods are used for tuning optimization hyperparameter, do they suffer from short-horizon bias (Wu et al., 2018)?

##########################################################################

I thank the authors for their responses. I carefully read them (revised manuscript) and also read other reviewers' feedback. The reviewers clarified some of my misunderstandings. However, I would like to remain at the current rating because my original concern about the novelty remains the same.

---

> ### Author Response · Authors · 2020-11-18
> **Response to reviewer 1**
>
> We thank the reviewer for the constructive feedback. We appreciate that the reviewer acknowledges the simplicity of the method and they find the empirical results convincing and supporting our claims. We also agree that releasing our code can potentially help the research community. All minor notes are fixed in the updated version of the paper.
>
> Answers to questions:
>
> 1. In terms of memory, asynchronous scheduling does not require any more memory than synchronous scheduling. Arguably the major limitation of asynchronous scheduling compared to synchronous scheduling is the additional implementation complexity (e.g worker / master node communication). We therefore provide an open-source implementation of our method along with the paper in AutoGluon, which enables other researchers to build new methods on-top or to use it for their applications.
>
> 2. Could the reviewer elaborate on the question? It is not possible to quantify this, since the integral is intractable, which is why we need to resort to MC estimation. As for their second question, it doesn't affect the model, since we're not updating the GP hyperparameters based on fantasy values.
>
> 3. This is indeed an interesting question, but beyond the scope of the current work. We are currently investigating it and are developing a method that “plans ahead” to take these downstream effects into account.
>
> 4. Actually the dashed lines in the plots in Figure 4 bottom right represent the performance of ASHA. We will clarify this in the caption.
>
> 5. Could the reviewer clarify in which plot exactly the validation regret decreases? Keep in mind that NASBench201 is based on an offline evaluated grid, which means 0 validation regret is possible on this benchmark. However, as argued in the main text, due to the intrinsic noise of neural networks training we don’t consider a regret < 10E-3 as significant.
>
> 6. We found Mobster to not be sensitive to its meta-hyperparameters (see for example the choice of kernel in Section 4.1). Since Mobster relies on a GP model it scales with the number of dimensions in the same way as any other Gaussian process based BO method, such as Dragonfly, for example.

---

### Official Review · AnonReviewer2 · 2020-10-29
**Model-based Asynchronous Hyperparameter and Neural Architecture Search combines asynchronous parallelism with GP-based BO for HNAS applications**

**Rating:** 6
**Confidence:** 4

**Review:**

The paper "Model-based Asynchronous Hyperparameter and Neural Architecture Search" combines asynchronous parallelism with GP-based BO for HNAS applications. In general, I don't see any strong novelty in this paper (see below), however, a few different techniques have been combined together to solve the important HNAS problem, which is relevant to the ICLR audience. The experimental section is valuable because it combines a number of well-thought benchmarks and a variety of SOTA algorithms are compared.

This paper may be better placed in a journal venue as a consolidation of techniques and through experimental evidence.

There is one main algorithmic contribution in this paper and an extensive and well-thought experimental section. The contribution is the use of GPs instead of TPE in BOHB. This is achieved using a classic multi-fidelity approach where the kernel is augmented with the fidelity r, which is, an incremental algorithmic improvement to a well-established algorithm. However, from a practitioner's perspective, it is interesting to see confirmation that this works well as expected. Once that the GP probabilistic model is introduced then it is possible to combine that with fantasies for batch BO and use two known approaches for asynchronicity.

Contribution 1 in the text (end of the first section) is not a contribution, I would suggest that the authors remove it from the contribution list. It is just a background section.

The paper is pretty hard to follow (even for a reader that is fairly acquainted with the topic). The writing feels rushed, there are several typos and imprecision in reporting the results, see my minor remarks at the end. The text should be reworked and made clearer, with smoother transitions between the subtopics, and paying a great deal of attention to giving enough background on the combination of techniques used.

The authors define the immediate regret. But aren't the authors using simple regret in all the result plots?

The authors say: " For all classification datasets we cap the regret at a lower bound of 10−3 , which corresponds to a 0.1% difference in classification accuracy to the global optimum". It is my understanding that this is incorrect because the global optimum is unknown. Instead, since the authors are computing the immediate regret this corresponds to a 0.1% difference in classification accuracy to the best-observed validation error across all methods, runs, and time steps on this dataset.

Minor remarks:
- Page 1: this sentence is badly constructed "Although their difference is subtle, they lead to substantially behaviours in practice".
- Page 3: K is used without being introduced.
- Page 4: in figure 2 the authors can add a legend for the 3 configurations (and the 3 different colors).
- Page 5: "Appendix B.All experiments" -> "Appendix B. All experiments"
- Page 7: formatting of the titles of the different paragraphs is off, e.g., use of ":", ".".
- Page 7: figure 4 bottom right middle and right is missing in the legend which one is ASHA and which one is Mobster.
- Page 8: why is BOHB being called BOHB in one figure and S-BOHB in another? Better using the same notation.
- Page 8: Figure 5, the names of the benchmarks on the top of the figures are confusing. Isn't the same benchmark evaluated twice for the multi-instance and mono-instance cases?
- Figure 1:
1) the left side doesn't depict BOHB, it represents one bracket in BOHB. So, the name on the top of the figure should be representative of that.
2) the task legend on the right side of the two figures is somewhat confusing. If I understand correctly that means that you are running 30 evaluations and that the colors say about the number of the evaluation.
3) I guess at this point in the text the authors are not interested in giving actual results. However, it would be better if the authors state if the data is fake; if this is not the case, what is the dataset that they are using?
- Since this is an implementation paper, it is a pity that checkpointing is not implemented because "is difficult in the multi-node distributed context". That would save some compute and give a slightly faster convergence.
- The text in figures 3, 4, and 5 is too small. Figure 3 should include the name of the benchmark inside the figure not just in the caption.
- It would be useful to clarify what the authors mean by multi-instance settings for people that are less familiar with cloud services. Is it a distributed setting where the workers are independent instances (VMs) and they are assigned a computation by a master instance?
- How are ordinal variables encoded in the search space (e.g., batch size in the tabular dataset)?

---

> ### Author Response · Authors · 2020-11-18
> **Response to reviewer 2**
>
> We thank the reviewer for taking the time to review our paper, and appreciate that they find it relevant to the ICLR audience and that they value the experimental section, “because it combines a number of well-thought benchmarks and a variety of SOTA algorithms are compared.” All minor remarks (typos, etc) are addressed in the updated version of the paper. Next we address the raised concerns:
>
> Clarity: Thank you for pointing out that parts of the paper might be hard to follow. We improved the writing and clarified some confusing parts regarding brackets and rungs in Section 2 and 3.
>
> Incremental: Our contribution goes beyond just replacing TPE by GPs in BOHB. Compared to the synchronous setting of BOHB, we need to take pending evaluations into account, which in turn requires fantasizing. Unlike in synchronous approaches, this is highly nontrivial (please see expanded details in revised appendix). Additionally, BOHB cannot extrapolate across  multiple fidelities.
>
> Immediate regret: We refer to the immediate regret to highlight that we compute the difference between the best value observed so far, rather than the final achieved performance to the global optimum. This is indeed the same as the simple regret and we will clarify this further.

---

### Official Review · AnonReviewer5 · 2020-11-09
**Thorough experiments suggests an effective approach, but overall limited insight and novelty with respect to previous work**

**Rating:** 6
**Confidence:** 2

**Review:**

Summary
-------

Algorithms for hyperparameter optimization either do not maintain a
model over configurations or are synchronous. This work proposes an
extension of the asynchronous successive halving algorithm by
introducing a Gaussian Process model which maintains a distribution over
configurations and resource levels (referred to as rungs). Their
algorithm, MOBSTER, is evaluated thoroughly against many baselines and
datasets.

Decision
--------

Overall, I am leaning towards marginally below acceptance threshold for
this paper. I do not have much confidence, due to my unfamiliarity with
this area. Your experiments section seems comprehensive and MOBSTER
generally outperforms commonly used algorithms for hyperparameter
optimization. Despite connections to ASHA and the median rule, there is
limited novelty and extension of prior work.

Originality
-----------

The way the work is presented suggests that the proposed approach is not
very novel. Your paper does not note novelty of the proposed approach as
a contribution (i.e. in Section 1.1). For someone unfamiliar with this
area, it would be good to discuss why this approach has not been
explored before despite its lack of novelty.

Quality and Clarity
-------------------

The paper is generally well-written. Some statements are vague or have
typos, and these can be found in the detailed/minor comments.

Strengths
---------

-   As someone not familiar with this literature, I think you do a good
    job at explaining previous work and your proposed approach. Section
    2 in particular, helps situate your work in the literature discussed
    in Section 1.2.
-   The experimental results are thorough and convincing. Although some
    choices remain unexplained, such as reporting regret and truncating
    performance, you include many baselines and evaluate performance on
    a wide variety of tasks. Some of the experiments on the larger
    models (LSTM, ResNet) seem inconclusive, but it's good to include
    them.

Weaknesses
----------

-   There is limited novelty in the proposed approach. This is not an
    issue on its own, but I think that the immediate challenges or
    drawbacks of the proposed approach are not adequately addressed.
    Perhaps it is not immediately straightforward to use a GP in
    combination with successive halving, but the challenges are not
    clear in Section 3. If it is straightforward, then it would be good
    to discuss the reasons why it wasn't done before.
-   Although it is noted that a main contribution is clarifying
    differences between ASHA and median rule, there is overall limited
    new insight. For example, much of Section 3 (on page 5) include
    statements on design choices for the experiments. These do not seem
    to be motivated from the proposed method, and generate little new
    insight and seem to be better suited to the experiment section.

Detailed Comments
-----------------

-   Abstract: "simultaneously reason across hyperparameters and resource
    levels, and supports decision-making in the presence of pending
    evaluations."

    This statement is confusing and doesn't convey what your approach
    is, or how it differs from existing methods.

-   Section 1: " \$**x**<sub>**\***</sub> ∈ arg
    min<sub>x∈\ mathcal{X}</sub> f(x)"

    You're discussing architecture and hyperparameters here, but this
    mathematical expression seems out of place. It is unclear if
    $\mathbf{x}$ is the hyperparameter or the input, because $f$ is
    defined later as a mapping for inputs $x$
    ($y_i = f(x_i) + \epsilon_i$) not hyperparameters.

-   Figure 2: I find the dashed lines fairly difficult to see. I
    appreciate the effort to label the difference between stopping and
    promotion, but I think the figure is actually not informative at all
    of what is going on. All of my understanding from this figure is
    from the caption.

-   Section 4: "We report the immediate regret…"

    The reasoning for this makes sense intuitively. However, can the
    results in the paper be directly compared to results of other
    papers? For example, under what conditions will the best performing
    algorithm with respect to immediate regret also correspond to the
    best performing algorithm with respect to validation error?

-   "For all classification datasets we cap the regret at a lower bound
    of 10−3, which corresponds to a 0.1% difference in classification
    accuracy to the global optimum."

    This seems like a bold choice. Are there similar decisions in other
    literature that can further justify this?

-   Figure 6: "After around 7000 seconds, MOBSTER’s model \[approaches
    the optimum\] much faster than ASHA"

    Is this accurate? Both methods seem to reach the optimum around the
    same time.

Minor Comments
--------------

-   Section 2, footnote: " Supporting pause-and-resume by checkpointing
    is difficult in the multi-node distributed context. It is simpler to
    start training from scratch whenever a configuration is promoted.
    This is done in our implementation"

    You should be clear here that pause-and-resume is done in your
    implementation. "This is done in our implementation" suggests that
    you do in fact start training from scratch whenever a configuration
    is promoted.

-   Section 4: "(for which the complexity considerable), which is why
    omitted them here."

    There seems to be some missing words here.

-   Scrolling and zooming in on the figures on pages 6/7/8 is very
    taxing on my relatively powerful desktop. I am not sure how the
    figures are embedded, but maybe these can be rasterized.

-   Figure 3: I assume RS is random search, but this is not said
    anywhere in the paper.

-   Figure 6: Why are the symbols (circle, diamond, squared) on the line
    plot missing in the figures here?

Post Discussion
--------------
After discussion, i have raised my score from 5 to 6.

---

> ### Author Response · Authors · 2020-11-18
> **Response to reviewer 5**
>
> We thank the reviewer for the constructive feedback. All minor comments are addressed in the updated version of the paper. We appreciate that the reviewer acknowledges the thorough and convincing experiments, and that we “do a good job at “explaining previous work and your proposed approach”. We should emphasize the significance of the improvement of MoBster over competitive state-of-the-art methods.
>
> Answers to the detailed comments:
>
> - Previous Bayesian optimization methods consider either the sequential multi-fidelity case, or the asynchronous parallel case, in isolation. Our method considers both, simultaneously, which turns out not to be straightforward. The only model-based method that considers multi-fidelity scheduling, but in the synchronous case, is BOHB. However, BOHB does not take pending evaluations into account, which is particularly critical in the asynchronous case; not doing so would lead to potentially attempting to evaluate the same (or very similar) candidates to the pending ones. In fact, in the distributed setting, BOHB or TPE just try to avoid sampling the same configuration multiple times by simply constraining the optimization of the acquisition function.
>
> - All hyperparameters and architectural choices are encoded in the vector x, and since we pass these directly as input to the GP model, we use the same notation. This is quite common in the AutoML literature.
>
> - Thanks for pointing this out. We will clarify this plot and make the dashed line more visible
>
> - In the case of NASBench201 results can be directly compared to results from other papers since the global optimum is known. For all other benchmarks we will release our code, so that other research can reproduce the direct numbers.  Note that in the deterministic setting, the best configuration with respect to the regret would also be the best configuration in terms of validation error. This does not necessarily hold in the noisy setting, however, since we use the best observed value across all methods and runs, the regret will be still strictly positive.
>
> - Training neural networks exhibit intrinsic noise caused by the mini-batch sampling and the weight initialization. We and others (see for example Ying et al) found this noise often around 10E-3 percent for the top performing configurations, which is why we used this lower bound to reduce clutter.
> NAS-Bench-101: Towards Reproducible Neural Architecture Search Chris Ying, Aaron Klein, Esteban Real, Eric Christiansen, Kevin Murphy, Frank Hutter ICML 2019
>
> - It is correct that both methods achieve the same final performance but MoBster converges already after 27000 seconds whereas ASHA after around 35000 seconds (a difference of ~2 hours).

---

> > ### Comment · AnonReviewer5 · 2020-11-23
> > **Brief response to this response and shared comments above**
> >
> >    Thank you for your detailed reply and revised paper. As a non-expert in this
> >    area, the paper was generally accessible. Your reply helped clear up some of
> >    my misunderstandings regarding relation to previous work and notation. I also
> >    appreciate your justification of empirical choices by pointing out relevant
> >    literature that supports these decisions.
> >
> >    The new details in Appendix A help demonstrate the technical novelty in the
> >    proposed approach. In my opinion, some of this should be worked into
> >    Section 3. However, it seems that R4 readily acknowledged the technical merit
> >    of the proposed method before these additional details. I will increase my
> >    score overall, but I am unable to increase my confidence.

---

### Author Response · Authors · 2020-11-18
**General comment to all reviewers**

First of all we thank all reviewers for taking the time to review our paper. We will upload the revised version of the paper by tomorrow. Before addressing each reviewer in turn, we would like to emphasize the following two aspects:

1. We are the first to combine GP-based BO with ASHA. Our main methodological contribution is the handling of asynchronicity with GP-based surrogates over different fidelities. While it is true that the individual parts are known from the literature, combining them in the asynchronous multi-fidelity setting is nontrivial and requires special attention to detail and additional algorithmic bookkeeping in the handling of the pending evaluations to obtain an efficient algorithm. Further, there is no natural way to handle pending evaluations with TPE (used inside BOHB), as it only estimates densities in the input space and is thus unequipped to predict the outcome of an pending evaluation.

2. We conduct a large-scale empirical evaluation of asynchronous scheduling. For example, we compare promotion-based and stopping-based ASHA, different kernels, successive halving vs Hyperband, etc. Our work shows that robustness in AutoML comes from the simplicity of judiciously combining rather primitive, but effective subroutines (e.g successive halving, fantasizing, etc.). Together with a fully-fledged open-source implementation, we believe our work will (a) enable practitioners to optimize their deep learning systems in a compute-efficient way, and (b) help the AutoML community to extend and develop new, even more efficient parallel HNAS on top of it.

---

> ### Comment · AnonReviewer4 · 2020-11-18
> **General comment to authors responses**
>
> Hello all, I have gone through other reviewers' comments and author responses, I pretty much appreciate the efforts made by the authors in general. Personally I don't have many technical issues, so next are really some high-level thoughts. By reading other reviews, I noticed that the major concern of almost all the other reviewers is either limited novelty or incremental result. First of all, I agree that it is not a paper with a brand new idea and I think the authors agree with that as well. However, although being completely natural, the coupling of different bricks does not seem to be trivial. I think the paper is worth some exposure to the public since it provides a thorough experimental support for some natural idea (which should be done by someone anyway), and could serve as a benchmark. So personally, I am still leaning toward accept, although it is probably, if accepted, not among the top batch of the overall accepted papers. I'm of course happy to hear from other reviewers on their opinions.

---

### Author Response · Authors · 2020-11-22
**Revised version of the paper is now online.**

After taking more time than anticipated, we finally uploaded a revised version of our paper, that addresses the points of the reviewers. More specifically, we changed the following points:

- To underscore the unique challenges that we have to face for pending evaluation and fantasizing in the asynchronous case, we added a detailed description in the Appendix
- To provide readers a more detailed picture of asynchronous multi-fidelity HNAS, we added pseudo code that outlines our algorithm to the appendix
- Typos and grammatical errors are fixed now. We also added missing details in the experiment section.
- We clarified confusing parts regarding rungs and brackets in Section 2 and 3.

---

### Decision · Program_Chairs · 2021-01-07
**Final Decision**

**Decision:**

Reject

**Comment:**

This paper is solid. It is correct, the text and author response demonstrate good knowledge of the area, the results are significant and solid, the experiments are strengthened by many independent runs (refreshing to see), the ablation study is well done, and the proposed distributed hyper-parameter and NAS alg is simple and practical. The paper is well written and reasonably polished.

The main drawback of the work in the eyes of the reviewers is that the paper is well described as a combination of existing ideas and a significant engineering effort with good but not stellar results. The reviewers found they did not gain any substantial technical insights from the work. As a result no reviewer was willing to champion the paper. However, the discussion, reviews, and author response made it clear that (1) the paper is enjoyable to read and informative, (2) the method is actually useful and performant, and (3) the combination of implementation details and methods is worth documenting. In balance, the paper is just below the bar. The program was extremely competitive this year.